# A general mechanism of ribosome dimerization revealed by single-particle cryo-electron microscopy

Linda E. Franken [1], Gert T. Oostergetel [1], Tjaard Pijning [1], Pranav Puri[2], Valentina Arkhipova[2], Egbert J. Boekema[1], Bert Poolman[2] & Albert Guskov [2]

Bacteria downregulate their ribosomal activity through dimerization of 70S ribosomes, yielding inactive 100S complexes. In *Escherichia coli*, dimerization is mediated by the hibernation promotion factor (HPF) and ribosome modulation factor. Here we report the cryo-electron microscopy study on 100S ribosomes from *Lactococcus lactis* and a dimerization mechanism involving a single protein: HPF[long]. The N-terminal domain of HPF[long] binds at the same site as HPF in *Escherichia coli* 100S ribosomes. Contrary to ribosome modulation factor, the C-terminal domain of HPF[long] binds exactly at the dimer interface. Furthermore, ribosomes from *Lactococcus lactis* do not undergo conformational changes in the 30S head domains upon binding of HPF[long], and the Shine–Dalgarno sequence and mRNA entrance tunnel remain accessible. Ribosome activity is blocked by HPF[long] due to the inhibition of mRNA recognition by the platform binding center. Phylogenetic analysis of HPF proteins suggests that HPF[long]-mediated dimerization is a widespread mechanism of ribosome hibernation in bacteria.

[1] Department of Biophysical Chemistry, Groningen Biomolecular Sciences and Biotechnology Institute, University of Groningen, Nijenborgh 7, 9747 AG Groningen, The Netherlands. [2] Department of Biochemistry, Groningen Biomolecular Sciences and Biotechnology Institute, University of Groningen, Nijenborgh 4, 9747 AG Groningen, The Netherlands. Correspondence and requests for materials should be addressed to A.G. (email: a.guskov@rug.nl)

Ribosomes are responsible for peptide-bond synthesis and are a main target for antibiotics[1]. They translate the genetic information from mRNA into a defined sequence of amino-acid residues, which eventually fold into mature proteins. When bacteria enter the stationary growth phase, the expression levels of hundreds of genes are changed and so is the synthesis of proteins[2–4]. The ability of a cell to control protein synthesis is essential for life. If ribosomal activity is not well controlled, over-activity or over-abundance of ribosomes can lead to serious problems, including stalling of ribosomes due to amino acid scarcity[5]. Ribosomes can be degraded to lower the translation activity but this is a costly strategy, both in terms of resources and competitive strength[5, 6]. A more efficient and faster response involves downregulation of ribosome activity.

Cells can do this via dimerization of 70S ribosomes into inactive 100S complexes[4], a process that has been observed in bacteria[7] and mammalian cells[8]. Ribosome dimerization is best documented for the Gram-negative bacteria *Escherichia coli*[5, 9, 10] and *Thermus thermophilus*[11]. Ribosome dimers of *E. coli*, connected through their 30S subunits, were first observed by electron microscopy in 1960[12], and proposed to be translationally inactive[13]. The mechanism in *E. coli* is mediated by the ribosome modulation factor (RMF)[10], which binds to a site in the 30S and interferes with the Shine–Dalgarno (SD) sequence. This prevents the interaction between mRNA and the 16S rRNA and leads to the formation of 90S dimers[11]. Subsequently, the hibernation promotion factor (HPF$^{short}$, YhbH) binds to a site that overlaps with that of mRNA, tRNA and initiation factors. Binding of HPF$^{short}$ modifies the structure of the 90S dimer to form the 100S complex[11]. Alternatively, a competing homolog of HPF$^{short}$, named YfiA (protein Y, pY or earlier termed RaiA or the ribosome-associated inhibitor A), inhibits ribosome activity by binding to the same location as HPF, but its C terminus protrudes into the binding site of RMF preventing 100S formation. Only the binding of RMF induces the conformational change in the 30S head domains, which has been proposed to directly take part in 100S formation[11]. Although *rmf* may be specific for γ-proteo-bacteria[14], most bacteria and some plant plastids[15, 16] carry a gene homologous to *hpf* $^{short}$, here referred to as *hpf* $^{long}$, and form 100S particles through a different mechanism, which we describe in this paper.

This second mechanism of dimerization exists in the majority of bacteria[7] and is mediated by a long version of HPF alone[6, 17–19]. This homolog of YfiA and HPF$^{short}$ is about twice as long and its C-terminal domain has no counterpart in *E. coli*. In a previous study[6], we demonstrated that deletion of the C-terminal part of this dual-domain protein (hereafter named HPF$^{long}$) in *Lactococcus lactis* results in the loss of dimerization. Thus, whereas the N-terminal domain may have a similar role and binding site as YfiA and HPF$^{short}$[20], the C-terminal domain of HPF$^{long}$ is necessary for dimer formation. Although the C-terminal domain differs from RMF, HPF$^{long}$ may induce a similar conformational change in the head region of the 30S subunit to facilitate the formation of 100S dimers[4, 6]. Alternatively, HPF$^{long}$ may directly trigger dimer formation upon binding at the interface, as has been suggested by Khusainov et al.[20]. To determine the binding site of HPF$^{long}$ and the actual mechanism of HPF$^{long}$-mediated dimerization, higher resolution structures are required[6, 7, 20], which we provide in this study.

We obtained by single particle cryo-transmission electron microscopy (cryo-TEM) two overall 100S maps of *L. lactis* ribosomes in different conformations at 19 Å resolution, which show a distinct rotational freedom of movement of 55° around the interface. We furthermore obtained a density-map of the *L. lactis* 70S ribosome as part of the dimeric complex at 5.6 Å resolution, and were able to confidently align two copies of the

ribosome to reconstruct the 100S structure thus significantly improving the density at the interface. Using the recently published structure of the 70S ribosome of the Gram-positive bacterium *Bacillus subtilis* (PDB-code 3J9W[21]) as a starting model, as well as the highest resolution *E. coli* 70S structure (PDB-code 4YBB[22]), we were able to model the structure of the *L. lactis* 70S ribosome. Moreover, we located both the N- and C-terminal domains of HPF$^{long}$ in the 100S ribosome as well as the interaction sites of the 70S particles within the dimer. We now present a second mechanism of ribosome dimerization that may be widely used in the bacterial kingdom and is distinctly different from the one proposed for *E. coli*.

## Results

**EM maps of the entire 100S dimer.** The dimerization interface of 100S structures is formed via interactions between the two 30S subunits. After a classification of 100S dimers into 32 classes, 62,499 of 163,121 particles were present in nine classes with a resolution better than 30 Å. Three different conformations were found (Supplementary Fig. 1). After aligning the bottom halves of the 100S ribosomes, a distinct change in position of the L1 stalk can be observed. The three conformations indicate a rotation of up to 55° around the 30S–30S interface. Also, the distance between the two ribosome monomers with respect to the interface increases from panel a to c; the different states are termed closed, intermediate, and open. The position of the L1 stalk does not change between particles of the most dominant closed state (59% of particles), and, contrary to what has been reported for dimers from *E. coli*[9], those dimers show an unexpected lack of flexibility around the interface, despite the very narrow dimer connection. In contrast, in particles belonging to the open conformation (35%), the position of the L1 stalk is displaced perpendicular to the interface, demonstrating an increased mobility, with respect to the closed state.

**EM map of a monomeric ribosome within the dimer.** Due to the conformational heterogeneity of the dimeric ribosomes, we used masks to refine the 30 and 50S subunits separately. A mask around the 30S subunit of one of the monomers yielded a density map at 5.9 Å (data not shown), and masking the 50S subunit of one of the monomers (Supplementary Fig. 2), yielded a density map at 5.6 Å resolution (Supplementary Figs. 3–5). The 5.9 Å (30S-masked) density map confirms the 5.6 Å (50S-masked) map. Therefore only the latter was used for modeling. The resolution is the highest in the center of the 50S subunit and follows a gradient through the 30S and toward the interface (Supplementary Fig. 3). A lower resolution for the 30S subunit is common in EM maps of ribosomes and is attributed to movement and flexibility within this part of the ribosome.

**Model of the 70S ribosome.** The genome of *L. lactis* (subsp. *cremoris*, MG1363) (NCBI Reference Sequence: NC_009004.1) encodes 20 and 32 proteins of the 30 and 50S subunit, respectively. Compared to *B. subtilis*, *L. lactis* lacks proteins L7a and bL25, and proteins uS14 and bL31 are present only as a single variant. The amino-acid conservation in ribosomal proteins is high. Along with the rRNA, we successfully assigned most of the ribosomal proteins (Fig. 1; Supplementary Fig. 6; Table 1) except uL1, L7/ L12, bL9, uL10, and uL11. Ribosomal protein bL9 may have been lost, due to a loose association with the ribosome[18]. The other missing proteins are known to be very flexible and, similar to the *Staphylococcus aureus* ribosome, lack sufficient density for modeling[20]. Two genes of *L. lactis*, *llmg 0899* and *llmg 2078*, have been annotated as encoding ribosomal protein uS15. However, a BLAST search with the sequence alignment of

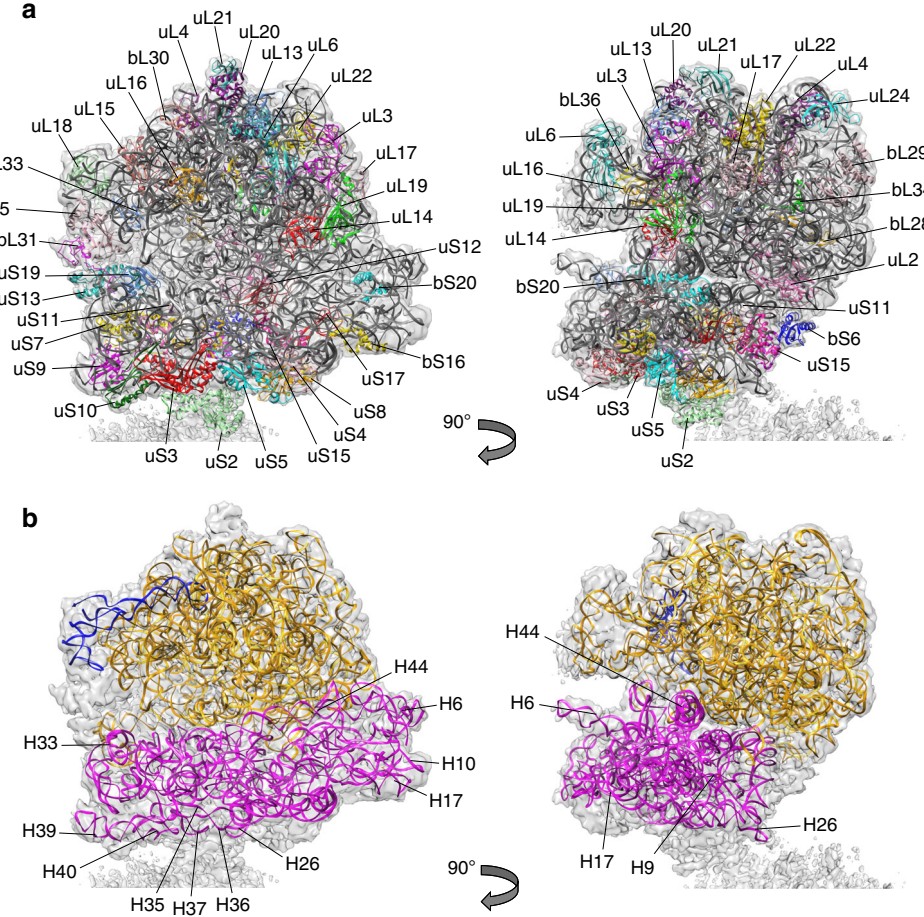

**Fig. 1** The complete 70S model of *L. lactis*. **a** Protein models fitted into the EM density and rotated over 90° with RNA in *gray* and proteins colored differently. Protein names are based on ref. [3]. **b** rRNA models fitted into the EM density and rotated over 90°

their products revealed that only *llmg 0899* truly encodes protein uS15. Additionally, we assigned protein bS21 in our map, which is absent in the model of *B. subtilis* ribosome[21]. For most of the protein subunits, the quality of the density was sufficient to trace the main chain, except for a few flexible loops and termini. Although at this resolution it is not possible to unambiguously assign amino-acid side chains, the quality of the map allowed their tentative assignment. The high sequence similarity between ribosomal proteins from *L. lactis* (PDB-code 5MYJ) and *B. subtilis* (PDB-code 3J9W) is reflected in their structural conservation, with the overall rmsd values for their 50 and 30S subunits of 1.65 and 2.2 Å, respectively.

**Model of HPF^long.** In order to improve the density of the map at the dimer interface a 100S map was reconstructed (Supplementary Fig. 7). After localization of all known ribosomal protein subunits, two regions of undescribed density remained, which we assign to HPF^long (Fig. 2). One density is located at the interface between the 30 and 50S subunits, and the other at the interface between both 30S subunits. This assignment was confirmed by the high similarity between the map densities and the respective homologous structures: YfiA from *E. coli* bound to the 70S ribosome of *T. thermophilus* (PDB-code 4V8I) and the C-terminal domain of ribosome-associated protein Y (PSrp-1) from *Clostridium acetobutylicum* (PDB-code 3KA5; Supplementary Fig. 8).

Structure prediction of HPF^long by the Predict Protein Server[23] (Supplementary Fig. 9) revealed a two-domain architecture

connected through an unstructured, solvent-exposed linker. The HPF^long N-terminal domain (residues 1-102, Fig. 2b), located between the 30 and 50S subunits, is in close proximity to proteins uS7 and uS12 and interacts with the 16S RNA at several locations (Fig. 2e). The C-terminal domain is located at the center of the 30S–30S interface (Fig. 2c). In the PSrp-1 dimer, the interface is formed by two hydrophobic patches at the surface of a β-sheet of each monomer (three Val, one Phe, and one Ile residue on each monomer). In the *L. lactis* HPF^long homology model, these residues are conserved, except for a single Val to Thr substitution (Supplementary Fig. 10). Furthermore, the PSrp-1 dimer is stabilized by a parallel β-sheet formed via the interaction of the first β-strand of each monomer with the last β-strand of the other monomer. The same hydrophobic and β-strand dimer stabilization is predicted for *L. lactis* HPF^long (Supplementary Fig. 9). The C-terminal domain of HPF^long is thus a dimer stabilized by hydrophobic patches and intermolecular β-sheet formation.

The EM map shows almost no density for residues 103–128 that connect the N- and C-terminal domains of HPF^long, which is indicative of a high flexibility. This is in agreement with the secondary structure predictions and the lack of sequence conservation among the linker domains in HPF^long (Supplementary Figs. 9 and 10). The length of the linker is rather well conserved and typically between 24 and 34 residues in ~4600 sequences analyzed (vide infra). The first visible residue (Asp129) of the C-terminal domain of HPF^long is at a distance of either 40 or 74 Å from residue 102, depending on which of the two copies of the protein is considered (Fig. 2f). We cannot

**Table 1 Data collection, refinement, and validation statistics**

| | |
|---|---|
| *Data collection* | |
| Microscope | Titan KRIOS with Cs-corrector |
| Camera | Falcon 2 |
| Voltage | 300 kV |
| Pixel size (Å) | 1.105 |
| Total dose (e⁻Å⁻²) | 25 |
| Micrographs collected (#) | 5275 |
| | |
| *Refinement* | |
| Total number of particles (#) | 163,121 |
| Particles in 50S masked map (#) | 43,530 |
| Resolution (Å; at FSC$^a$ = 0.143) | 5.6 |
| CC$^a$ (model to map fit$^b$) | 0.800 |
| | |
| *RMS$^a$ deviations* | |
| Bonds (Å) | 0.008 |
| Angles (°) | 1.262 |
| Chirality (°) | 0.06 |
| Planarity (°) | 0.008 |
| | |
| *Validation$^c$* | |
| Clash-score$^d$ | 18.69 |
| | |
| *Proteins* | |
| MolProbity score | 2.55 |
| Favored rotamers | 91.3% |
| Ramachandran favored | 82.31% |
| Ramachandran allowed | 17.54% |
| Ramachandran outliers | 0.15% |
| | |
| *RNA* | |
| Correct sugar puckers | 99.49% |

$^a$Fourier shell correlation; correlation coefficient; root-mean square
$^b$Only across atoms in the model; as indicated in Phenix[47]
$^c$As indicated by MolProbity[52]
$^d$Number of steric overlaps (>0.4 Å) per 1000 atoms

unambiguously determine which route the connecting loop follows, but the maximum reaching distance (99 Å for 26 residues, assuming an average Cα–Cα distance of 3.8 Å per residue) suggests that the long route for spanning two opposing ribosomes is well possible.

**Dimer interface.** Each ribosome in the 100S particle interacts at the dimer interface with both copies of the HPF$^{long}$ C-terminal domain (Fig. 2c, d). From the perspective of one 70S ribosome in the dimer, the main interaction is between its uS2 protein and copy A (randomly assigned) of the HPF$^{long}$ C-terminal domains; whereas RNA helix 40 and most likely the N terminus of bS18, interact with the copy B. Chain bS18 also interacts with RNA helix 26, which seems to be involved in the dimerization through interactions with helix 26 of the opposing ribosome (at least in *L. lactis*) (Fig. 2c).

**Rotation in the 100S ribosomes.** The model of the 70S ribosome can be used to compare three conformations of 100S ribosomes (Supplementary Fig. 1). Two copies of the 70S model without the C-terminal domains of HPF$^{long}$ were rigid-body fitted in each map (Fig. 3). The 55° rotational difference manifests in a change of the orientation of the helices 26 of the 16S RNAs from facing each other in the closed state to interacting with protein uS2 of the opposing ribosome in the open state. The intermediate state has a lower resolution and a less defined interface, although its well-defined L1 Stalk points to a stable conformation. The fit of the two models in the intermediate map indicates that the two RNA helices 26 are interacting in this conformation; the density

for HPF$^{long}$ seems to be (partially) lost. Whether ribosome dimers assemble into these different conformations or are able to switch after assembly remains to be studied.

When assigning the two C-terminal domains of the HPF$^{long}$ dimer each to a specific ribosome, it was not possible to account for the two most rotated dimer conformations (Supplementary Fig. 1a, c); the C-terminal dimer was torn apart or crashed into itself. Both 100S conformations can only be accounted for when the C-terminal domains are fitted together as an intact dimer. The best fit is obtained by moving HPF$^{long}$ such that the two-fold symmetry axes of the 100S and HPF$^{long}$ coincide again. From the closed (Supplementary Fig. 1a) to the open state (Supplementary Fig. 1c) the dimer shifts ~6 Å and rotates 27° over the long axis. This compensates the increased distance between the two 70S ribosomes and the rotation. Most likely, the HPF$^{long}$ C-terminal domains are not flexible and maintain their dimer interface, while the flexibility in protein uS2 allows the rotation.

**Oligomeric state of HPF$^{long}$ in solution.** We performed size-exclusion chromatography coupled to multi-angle laser light scattering (SEC-MALLS) measurements to determine the oligomeric state of HPF$^{long}$ in solution. Purified HPF$^{long}$ in the native state has a molecular mass of ~40.5 kDa, corresponding to about twice the calculated mass on the basis of the amino-acid sequence (21.3 kDa). Thus, HPF$^{long}$ forms a stable dimer in solution (Fig. 4).

**Distribution and phylogenetics of HPF$^{long}$.** A BLASTp search for HPF$^{long}$ homologs yielded about 7000 plausible hits with at least 30% sequence identity. Sorting by sequence coverage showed a steep drop from full-length (~185 residues) to half-size proteins after about 4600 hits. HPF$^{long}$ homologs were present in almost all bacterial phyla, but not in Gram-positive *Acholeplasma sp.*, Gram-negative *Borrelia sp.*, Euryachaeota, and Crenarchaeota.

Alignment of a selected set of 110 homologs (Supplementary Fig. 10) revealed that 95 sequences cover all three regions of HPF$^{long}$, and they align well with both N- and C-terminal domains; 15 sequences covered only the N-terminal domain. In the 'full-length' homologs, the linker between the two domains showed very little conservation in sequence and composition, and its length varied between 16 and 62 residues; the majority being between 24 and 34 residues. In the C-terminal domain, the five hydrophobic residues at the dimer interface are highly conserved: three out of five are (almost) invariantly present, while for the remaining two a hydrophobic substitution is typically present (e.g., valine to isoleucine) (Supplementary Fig. 10).

A phylogenetic tree constructed for the 110 sequences (Fig. 5) revealed that the distribution and evolutionary relationship of HPF$^{long}$ homologs is reflected in the distribution of bacterial phyla. *L. lactis* HPF$^{long}$ appears in a clade representing (almost) all Gram-positive, low G+C phyla at one 'end' of the phylogenetic tree. This clade (with the exception of one *S. aureus* species) contains full-length HPF$^{long}$ homologs with the inter-domain linker length comparable to that of *L. lactis* (31 residues). In the Gram-positive, high G+C actinobacteria at the other 'end' of the phylogenetic distribution, the linker is significantly longer. The shortest linkers are observed in the sequences from Thermodesulfobacteria. Gram-negative bacterial phyla are situated in the 'middle' of the phylogenetic tree. Notably, the β- and γ-proteobacteria (*Neisseria* and *Escherichia* species) only contain short YfiA homologs (originally described as HPF and YfiA). Interestingly, RMF homologs were only found (by BLASTp) in γ-proteobacteria.

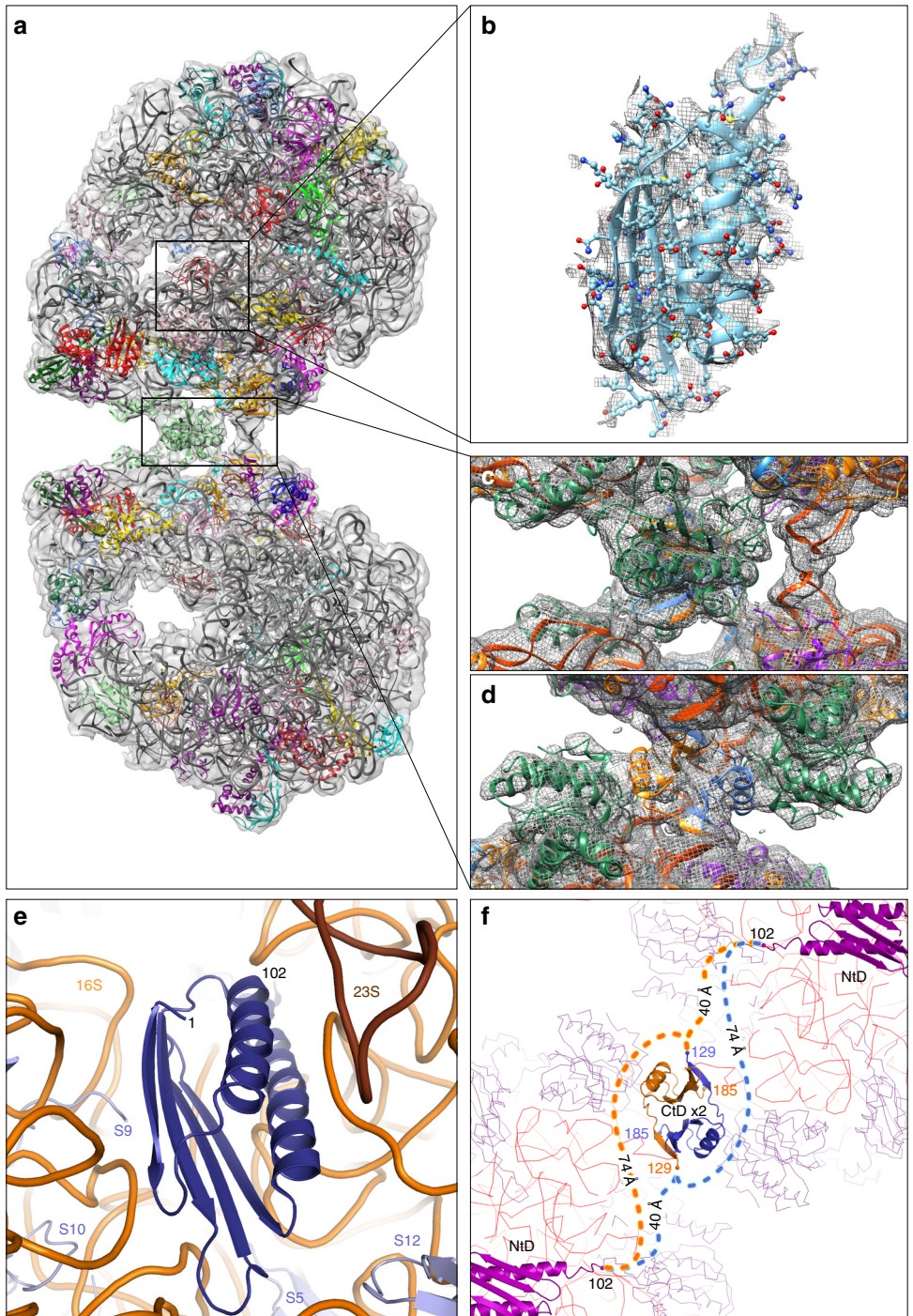

**Fig. 2** Location of the N- and C-terminal domain of HPF$^{long}$. The extra, unmodeled densities in the map, indicated by the *boxes* of **a**, correspond to HPF$^{long}$ N-terminal domain (**b** and Supplementary Movie 1) and C-terminal domain (**c**, **d** and in Supplementary Movie 2; colored *gold* and *blue*). **d** has been rotated 90° with respect to **a**, **c**, showing the dimeric interaction of the C-terminal domains. **e** The surrounding proteins (*steel blue*) and RNA (*orange*) of the N-terminal domain. **f** The linking of the distant N- and C-terminal domains (NtD and CtD) is shown schematically. Their respective locations in the dimer are clarified by the surrounding ribosomal proteins (*purple lines*) and 16S rRNA (*red lines*)

## Discussion

The here-reported structure of the *L. lactis* 70S ribosome is that of the fifth bacterial species. Structural variations in ribosomes have been reviewed by Khusainov et al.[20]; and some of them (e.g., variations in the lengths of rRNA loops) could point to the involvement of rRNA extensions in translation control in a species-specific manner. Most knowledge on initiation, translation, hibernation, mRNA recognition, and ribosome management is based on studies in *E. coli*. The lack of (structural) studies on ribosomes across the bacterial kingdom is one of the most important shortcomings in understanding vital and clinically relevant species-specific regulation of translation and phenomena such as antibiotic resistance[20].

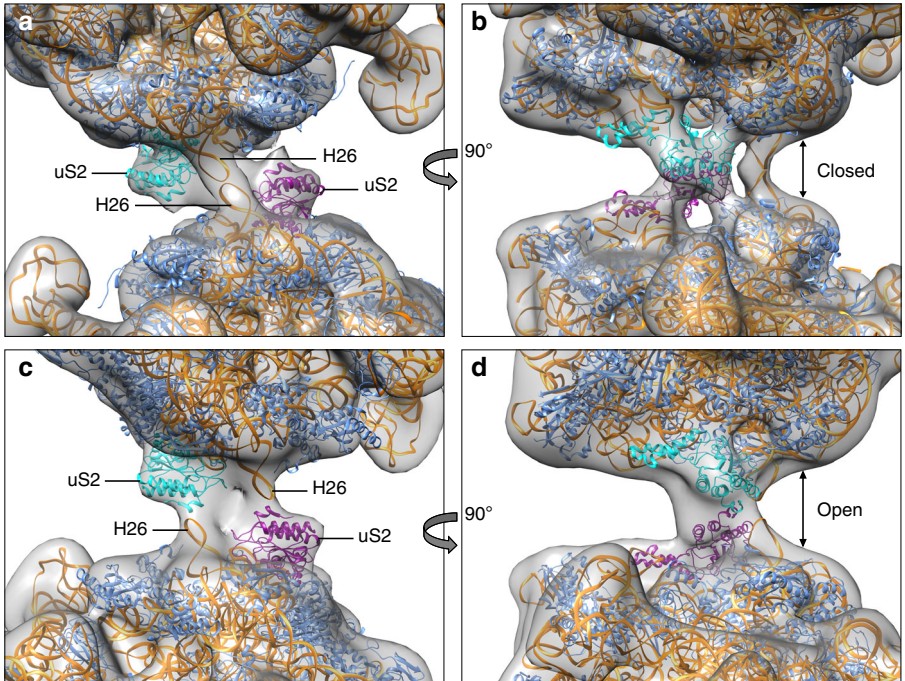

**Fig. 3** Different conformational states of 100S ribosomes. Rigid-body fits of two 70S models without the C-terminal domains of HPF[long] into the low-resolution maps of the two most extreme conformations, closed state (**a**, **b**) and open state (**c**, **d**). The difference between the two classes depicts a 55° rotation around the interface. The 'closed' conformation closely resembles our 5.6 Å map. Colors indicate 16S rRNA (*orange*), protein uS2 (*cyan* and *purple*) and other proteins (*blue*). The rotation causes H26 to change its interaction from H26 to protein uS2 from the other ribosome, effectively widening the space between the two 70S ribosomes within the dimer (*arrows* in **b**, **d**)

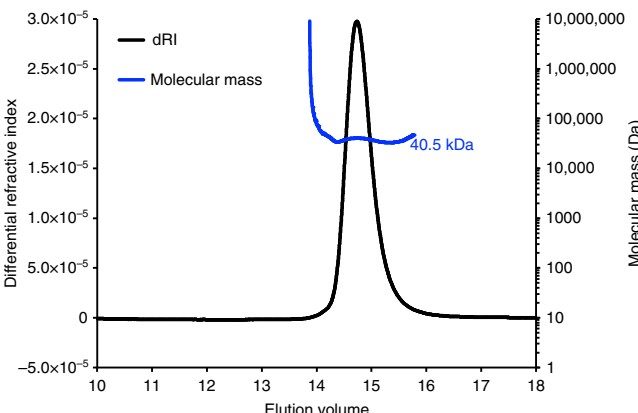

**Fig. 4** Oligomeric state analysis of HPF[long] by SEC-MALLS. The Superdex 200 10/300 column (GE Healthcare) was equilibrated with 100 mM Tris-HCl, 150 mM NaCl (pH 8.0), and the protein was injected in the same buffer. The chromatogram (elution volume is indicated on the *x* axis) shows the readings of refractive index (RI) detector in *black* (the scale for the RI detector is shown in the *left-hand* axis). The *thick blue line* indicates the calculated molecular mass of the eluting protein throughout the chromatogram (scale on the *right-hand* axis). The calculated molecular weight is 40.5 kDa (molecular weight of a monomer is 21.3 kDa)

We report a novel mechanism of ribosome dimerization via a long-HPF-type two-domain protein. The N-terminal domain of HPF[long] is structurally conserved and has a similar function and binding site as YfiA and HPF[short] from *E. coli* (Supplementary Fig. 11). It blocks ribosome function by binding to the sites for tRNA, initiation factors IF1 and IF3, and elongation factor G[11], thereby inhibiting ribosome activity and preventing the dissociation of the 70S particles into the 30 and 50S subunits[24–26]. Analogous to HPF[short] from *E. coli*, the N-terminal domain of HPF[long] strengthens the association of the 30 and 50S into 70S ribosomes, which renders them less susceptible to endoribonuclease action and degradation[27].

In contrast to RMF from *E. coli*, the HPF[long] C-terminal domain does not interact with the SD-aSD sequence but with protein uS2 at the dimer interface. Protein uS2 is essential for ribosome function[28, 29] and is also part of the mRNA 'platform binding center' (PBC) along with proteins uS7 and uS11 and 16S rRNA helices 26 and 40[30, 31]. This platform has been proposed to be a common site dedicated to binding mRNAs prior to the actual translation initiation to increase control over translation[30]. Supplementary Fig. 12 clearly demonstrates the overlap between the *L. lactis* dimer interface and the locked mRNA structure. We therefore propose that the HPF[long] C-terminal domain blocks the PBC by acting as a general, rather than a specific, inhibitor of mRNA translation initiation.

We could not unambiguously connect the N- and C-terminal domains of HPF[long] in 100S ribosomes. The linker is sufficiently long to link each N-terminal domain to each C-terminal domain and allow the rotation over the dimer interface (Fig. 3). The shorter (~40 Å) and longer route (~74 Å) can be bridged by the linker in the majority of HPF[long] homologs. It is even possible that both routes are used, because HPF[long] is a natural dimer, strongly interacting via the two C-terminal domains. This is supported by (i) the finding that a dimeric HPF[long] fits both main conformations of the 100S ribosome (Fig. 3); (ii) the hydrophobic nature of the dimer interface; (iii) the fact that these hydrophobic residues are conserved in all proteins that carry the C-terminal domain (Supplementary Fig. 10); (iv) the X-ray crystal structures of two homologs (PDB-codes 3K2T and 3KA5, see Methods section); and (v) the SEC-MALLS experiments (Fig. 4).

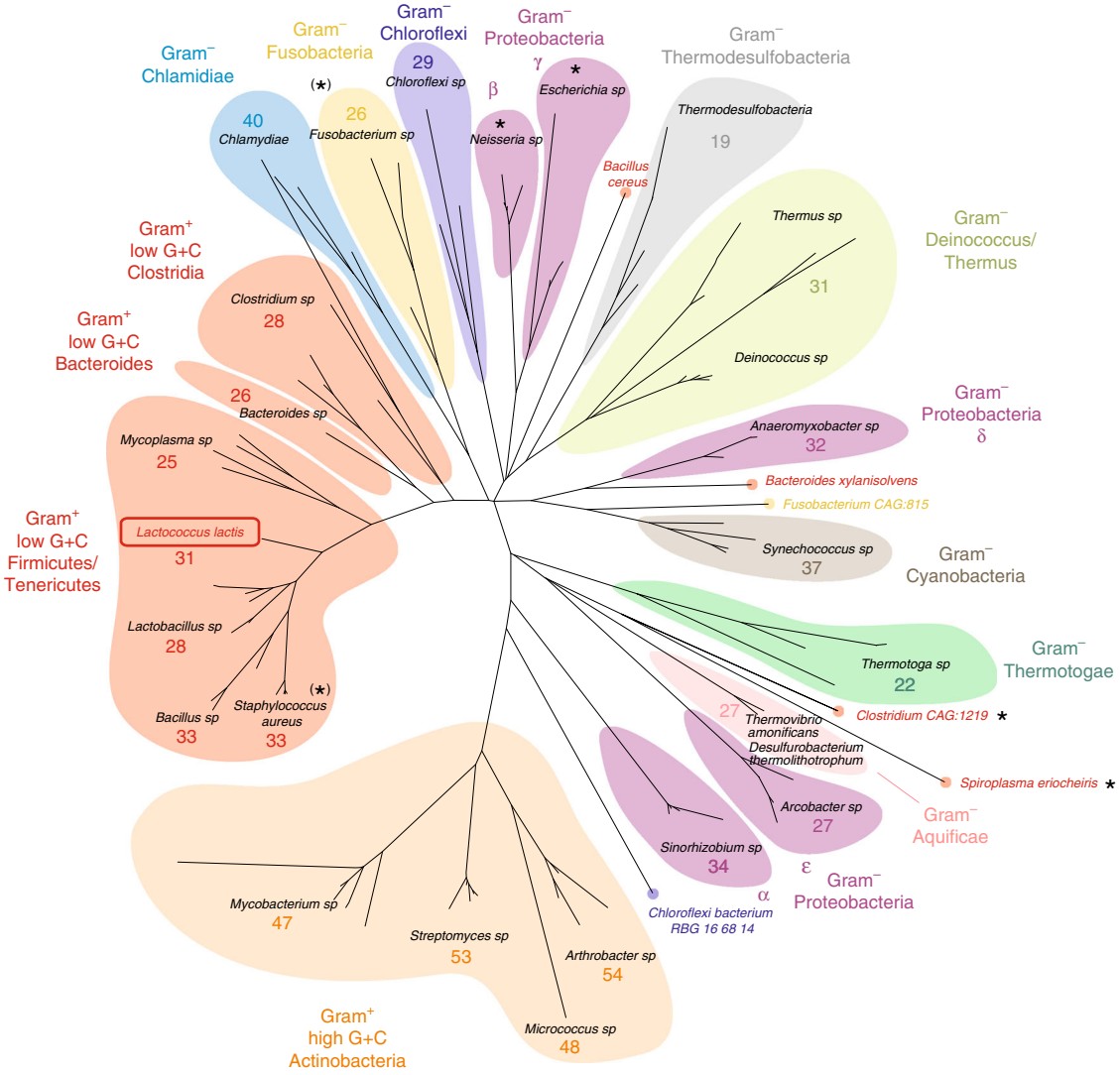

**Fig. 5** Phylogenetic tree of HPF homologs. The tree is based on 110 proteins homologous to HPF$^{long}$ from *L. lactis*. The average linker length of HPF$^{long}$ is given by the number of residues in each of the lineages

We compared the ribosome dimerization mechanism by HPF$^{long}$ in *L. lactis* with that of HPF$^{short}$ and RMF in *E.coli*[5, 9] (Fig. 6). We used the map of Kato et al.[9], because it is of higher resolution (~18 Å) and in better agreement with the structure compared to the map from Ortiz et al.[5] (Supplementary Fig. 13). Superposition of the maps from *E. coli* and *L. lactis* based on the bottom ribosomes reveals the enormous difference in overall orientation of the top ribosome. The first dimer interaction site between the two 30S head-domains[5, 11] is not present in *L. lactis*, and we propose that also in *E. coli* only one interaction site exists, in agreement with the map from Kato et al.[9], and the very weak and loose nature of the *E. coli* dimer interface[5, 9].

Despite the large difference in the overall 100S structures, the main protein involved in the interface of *E. coli* and *L. lactis* dimers is uS2. The actual binding site on uS2 is not conserved (Fig. 6c). In *E. coli*, this dimer interaction site comprises the binding of uS2 to uS3, uS4, and uS5 from two opposing ribosomes[5, 9, 11] (Supplementary Fig. 13). The uS3–uS5 proteins form the mRNA entrance tunnel, which is blocked by the dimerization. In *L. lactis*, uS2 binds to the C-terminal domain of HPF$^{long}$, which causes the actual dimerization, and proteins uS3–uS5 are not part of the interface.

In *E. coli*, HPF and YfiA are not completely able to prevent the separation of 30 and 50S for initiation[32]. Instead, they specifically hinder translation of leaderless mRNA[14], slowing down their translation, but also reducing translation errors[33]. RMF is essential to further block translation. It induces dimerization by a conformational change in the ribosome 30S head-domains. While the N-terminal domain of HPF$^{long}$ is analogous to YfiA and HPF$^{short}$, the binding-site of the C-terminal domain from *L. lactis* HPF$^{long}$ is not the same as that of RMF. Still, the HPF$^{long}$ C-terminal domain likely also completes the ribosome inactivation. The direct involvement of HPF$^{long}$ C-terminal domain at the dimer interface leaves the SD-aSD sequence accessible in the *L. lactis* ribosome dimers. Also, contrasting our previous analysis[6], there is no conformational change in *L. lactis* ribosomes upon binding of HPF$^{long}$ (Fig. 6e, f).

The existence of two different mechanisms of ribosome dimerization, and the conservation of protein uS2 as main interaction partner, supports the idea that dimerization has been rescued in γ-proteobacteria by the *rmf* gene product upon loss of the C-terminal domain of HPF$^{long}$. Given the fact that RMF only exists in γ-proteobacteria, our homology search confirms and extends the conclusion that the *E. coli* dimerization mechanism, albeit best studied, is in fact not the most widely spread strategy[14].

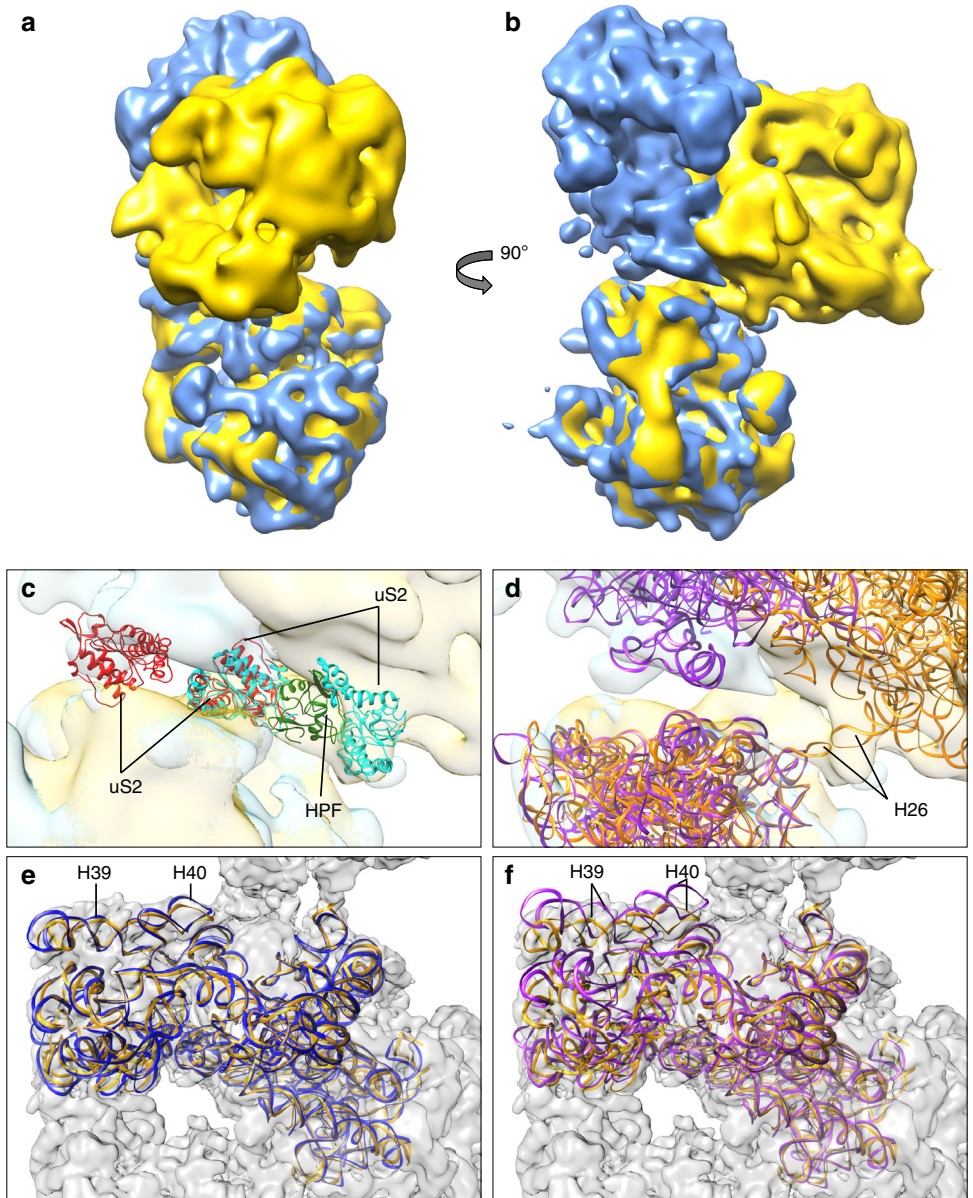

**Fig. 6** Ribosome dimerization: RMF vs. HPF[long]. **a–d** Comparison of 3D maps of *E. coli* (*blue*[9]) and *L. lactis* (*yellow*) 100S ribosomes. The density for the *E. coli* ribosome was obtained by taking the maximum voxel value of two copies of the EM-density map from Kato et al.[9]. Two copies of the *L. lactis* model and two copies of the *T. thermophilus* structure with RMF (PDB-code 4V8G), thus displaying the mechanism of *E. coli*, not *T. thermophilus*, were fitted into the corresponding density (**c**, **d**). **c** Zoom in of dimer interface showing proteins uS2 (marked *red* for *T. thermophilus* and *cyan* for *L. lactis*). In *L. lactis* the dimerization process is mediated by the C-terminal domain of HPF[long] (*green*), whereas in *T. thermophilus* uS2 interacts with proteins uS3, uS4, and uS5 from the opposing ribosome (previously described as second contact site[11]). **d** The RNA chains of the *L. lactis* 100S (*orange*) and *T. thermophilus* 100S structures (*purple*). Helix 26 plays a key role in dimerization of *L. lactis* ribosomes but not in *T. thermophilus*. **e**, **f** Comparison of the 16S rRNA of ribosomes from *L. lactis* (*orange*) to *T. thermophilus* in apo-state (**e**, *blue*) and RMF-bound state (**f**, *purple*). The 100S map from *L. lactis* is depicted in *gray* in the background. By looking at the helices 39 and 40 of the head domain, it is clear that the ribosome from *L. lactis* is in the apo-state and does not undergo a conformational change upon binding of HPF[long]

Full-length homologs of *L. lactis* HPF[long] were found in almost all bacterial phyla. Few lacked the N-terminal domain (e.g., some *M. tuberculosis*, *S. aureus sp.*) or even an HPF homolog at all (e.g., *Acholaplasma and Borrelia sp.* as well as Archaea); all *Neisseria sp.* lacked both a C-terminal domain as well as RMF in their genome. These organisms may lack or have alternative mechanisms of translation regulation and ribosome stabilization.

One of the key differences between ribosome hibernation in *L. lactis* and *E. coli* relates to the states of HPF[long] and RMF. While in *E. coli* two unoccupied ribosomes each collect one copy of RMF and then dimerize, HPF[long] is a dimer in solution and therefore it can in principle initially gather one, and then the other ribosome (Fig. 7). Our findings indicate that the 70S–70S interactions in *L. lactis* are due to HPF[long] exclusively. With increasing levels of HPF[long], more 100S ribosomes are formed but there will be an optimum. If HPF[long] is produced in excess of ribosomes, the equilibrium will shift from 100S to 70S-HPF[long], i.e., monomeric ribosomes with dimeric HPF[long] bound (Fig. 7). Both 100S and 70S-HPF[long] are inactive, but the dimeric state may protect against degradation by shielding protein uS2, which forms the main dimer interaction site in both *L. lactis* and *E. coli*. Next to the HPF[long]/ribosome ratio, the

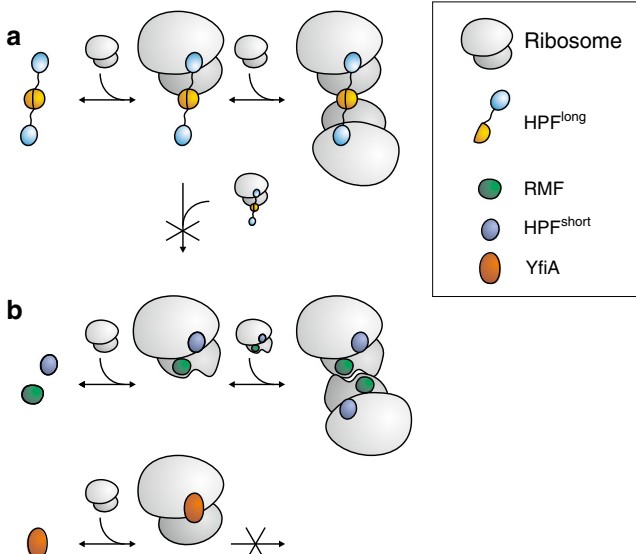

**Fig. 7** Schematic of dimerization of ribosomes in *L. lactis* and *E. coli*. Because HPF$^{long}$ is a dimer in solution it will collect two ribosomes consecutively (**a**). Prolonged exposure to an excess of HPF$^{long}$ shifts the equilibrium toward monomeric inactive ribosomes. In *E. coli*, (**b**) each ribosome first collects a copy of HPF$^{short}$ and RMF. The conformational change upon binding of RMF allows the two ribosomes to dimerize. When YfiA binds at the location of HPF$^{short}$, RMF can no longer bind and the ribosomes do not dimerize

presence of mRNA and the availability of amino acyl-tRNA (more abundant in the exponential than in the stationary phase) determine whether the ribosomes are monomeric or dimeric.

Ueta et al.[18] have proposed that in the exponential phase of growth HPF$^{long}$ associates with 100S but not with the 70S particles. The expression of HPF$^{long}$ increases at the onset of the stationary phase of growth and thus more dimers are formed. During prolonged stationary growth, the number of 70S ribosomes associated with HPF$^{long}$ increases and the number of dimers decreases. We now explain these observations by the fact that HPF$^{long}$ is a dimer. As occasionally a 100S particle gets reactivated, the balance shifts from 100 to 70S. Notably, in *E. coli*, HPF$^{short}$ and YfiA compete for the same binding site where YfiA, but not HPF$^{short}$, prevents binding of RMF. This can also lead to two populations of inactivated ribosomes, but in this case the 70S particles have YfiA bound and the 100S particles have HPF and RMF bound.

In this study, we present a new HPF$^{long}$-mediated mechanism of ribosome hibernation, which may be present in nearly all bacterial species and shows stark differences to the RMF-induced dimerization in γ-proteobacteria. HPF$^{long}$ is a dual domain protein that forms a dimer in solution. Although the N-terminal domain of HPF$^{long}$ binds in the same location as YfiA and HPF$^{short}$ from *E. coli*, the location of the C-terminal domain does not overlap with that of RMF. While RMF induces a conformational change to allow dimerization, the dimeric C-terminal domains of HPF$^{long}$ directly form the dimer, to which the two ribosomes, remaining in apo-state, are bound. In *E. coli*, ribosomal activity is inhibited by dimer formation, which blocks the mRNA entrance tunnel, and by RMF itself, which interacts with the SD-aSD sequence. In contrast, in *L. lactis* the SD-aSD and mRNA tunnel are still accessible, and ribosome activity is hindered by the C-terminal domain of HPF$^{long}$, which blocks the PBC. Interestingly, both species are able to shift ribosome hibernation from a dimeric inactive to a monomeric inactive state. In *E. coli*, this can be realised by different expression levels for YfiA and HPF$^{short}$, while in *L. lactis* HPF$^{long}$ alone is

responsible. Thus, our study gives an example of how vital functions in bacterial physiology can evolve through the acquisition of different molecular mechanisms in different species.

## Methods

**Cell growth and purification.** Cells were grown and ribosomes collected as described previously[6]. In short: *L. lactis* MG1363 strain were grown as standing cultures in M17 medium (Difco Laboratories, Detroit, MA, USA), supplemented with 0.5% (w/v) glucose (GM17); 5 µg ml$^{-1}$ chloramphenicol and/or 2.5 µg ml$^{-1}$ erythromycin were added when cells were transformed with the required plasmid(s).

*L. lactis* cells were collected by centrifugation (7000×*g*, 10 min, 4 °C) after 3 h (logarithmic growth phase, 70S fraction) or 7 h (stationary growth phase, 100S fraction). The cell pellet was resuspended in buffer I (20 mM Tris-HCl, pH 7.6), 15 mM magnesium acetate, 100 mM ammonium acetate, and 6 mM 2-mercaptoethanol containing 1 mM phenylmethylsulphonyl fluoride (PMSF), and lysed by vortexing with 0.2 mg glass beads in an ice-cold Tissue lyser (Qiagen, Venlo, The Netherlands). The homogenate was centrifuged (9000×*g*, 15 min, 4 °C), the supernatant was collected and the pellet resuspended in buffer I supplemented with 1 mM PMSF. The suspension was centrifuged again and the combined supernatants were layered onto a 30% sucrose cushion in buffer I and centrifuged (206,000×*g* for 3 h at 4 °C). Resuspension of the pellet in buffer I gave a crude ribosome preparation.

To further purify ribosome monomers and dimers, 400 µl of ribosome preparation was loaded onto a linear 10–40% 12 ml sucrose density gradient column in buffer I and centrifuged in a SW 32.1 Ti rotor (Beckman) at 125,000×*g* for 80 min at 4 °C. The gradient was divided into 400 µl fractions after which their 260 nm absorbance was measured using a UV-1700 CARY bio UV-Visible Spectrophotometer (Agilent Technologies, Palo Alto, CA, USA). The fractions containing the ribosome monomers (70S) or dimers (100S) were dialyzed at 4 °C against buffer I and prepared for electron microscopy.

**Electron microscopy.** A volume of 3 µl of ribosome sample of either 100- or 70S ribosomes was pipetted onto glow-discharged 400 mesh holey carbon grids (Quantifoil, 2/2) and plunge-frozen in liquid ethane with a Vitrobot (FEI, Eindhoven, The Netherlands). All images were recorded on a Titan KRIOS (FEI) equipped with a high brightness field-emission gun operated at 300 kV, a Cs-corrector (CEOS, Heidelberg, Germany) and a Falcon 2 DED camera (FEI) at the NeCEN (Leiden, The Netherlands). Images were taken automatically with EPU software (FEI) using a dose of 25 e$^-$ per Å$^2$ and focus settings of 1000, 1600, and 2500 nm defocus with a final pixel size of 1.105 Å. No movie-frames were recorded.

**Data processing.** From the raw images, the purified 70S peak-fraction micrographs seemed to contain as many dimers as the 100S peak. Likely, HPF$^{long}$ dimers were co-purified with monomeric ribosomes in the 70S fraction[18, 34]. During 2D and 3D processing, 100S particles from cells in the exponential and stationary phase of growth did not separate into different classes, but mixed in equal amounts. Therefore both data sets were combined to increase the amount of data.

CTF parameters were estimated using CTFFIND4[35]. The resulting _avrot.txt files were used to select micrographs with a CTF fit better than 7 Å, by applying a 0.1 threshold for the drop in the cross-correlation between calculated fit and data. This yielded a total of 2713 micrographs of the original 100S fraction and an additional 2562 micrographs of the 70S fraction.

RELION 1.3 and 1.4[36] were used for all processing apart from the initial model. Due to the lack of mass in the center of the dimers, automated particle picking resulted in uncentered picks. On top of that, the high similarity between 70 and 100S particles, which were both present in all micrographs, rendered 2D and 3D cleaning insufficient. To avoid the presence of monomeric particles in the dimeric structure, particle picking was done manually. This was done in RELION 1.3 for the 2500 nm defocus images and in Xmipp 3.0[37] for the micrographs that were taken with less defocus, making use of contrast enhancement and band-pass filtering.

Initially, particles were down-sized four times and the dataset was cleaned using reference-free 2D classification in RELION. To obtain an initial model, the last 2D class-sums were exported into Xmipp 3.0 to run two rounds of RANSAC[38] feeding the best result of the first as a model for the second round after which ten very similar models were created, one of which served then as initial model in RELION 3.1 for the first 3D classifications. All references were low-pass filtered to 60 Å to avoid a strong influence of the reference on the results.

Visualization and comparison of 3D models were done using Chimera[39]. Volume alignments based on one half of the dimer were obtained by applying the segmentation tool[40] and grouping the segments until two monomeric maps remained. The freedom of rotation within the dimer was measured taking the two most extreme conformations and first fitting one class on the bottom segment of the other using the fit in volume tool. Secondly, that same class was fitted onto the top segment. The tool gives the applied rotation.

The clean set contained 62,499 particles and was further processed with unbinned pixels. To compensate the movement, the next classifications and refinement used several soft-edged masks (Supplementary Fig. 2). Simple circular

masks were created in Xmipp 3.0 and tighter masks were created in RELION 1.3 based on sub-volumes that were created using the segmentation tool in Chimera. The final masked map totaled 43,530 particles (28,747 originating from the 100S and 14,782 from the 70S fraction) and was refined in RELION 1.4 based on a 50S mask, while the auto-mask function of RELION 1.4 was used to create the final masks for post-processing to allow for a tighter mask and obtain information beyond the 50S mask. Post-processing was done in RELION 1.4, correcting the map for the modulation transfer function of the detector. For sharpening, two ad-hoc b-factors were used: −150 and −100. To limit the sharpening of noise in the 100S interface, the second, more conservative value was used prior to reconstructing the 100S map from two copies of the map (vide infra). Reported resolutions are based on the gold-standard FSC = 0.143 criterion[41] and local resolutions were determined in Resmap[42]. Since the final map from the 50S mask (5.6 Å) and 30S mask (5.9 Å) were in agreement, all modeling was based on the 50S map.

The 100S density map was reconstructed from two copies of the 5.6 Å resolution map (Supplementary Figs. 3 and 7), which were first fitted into the best 100S class (19 Å resolution, Supplementary Fig. 1a) and then into each other using the Fit Volume tool in Chimera. The single 100S was created by taking the maximum voxel value between the two aligned maps, getting for each voxel the largest value. This two-fold symmetrical 100S map with improved density at the interface was only used to model the interface.

**Modeling.** The structural modeling was started by rigid-body fitting the B. subtilis structure (PDB-code 3J9W[21]) in the density map in COOT[43]. RNA sequences for L. lactis were taken from the RNAcentral database (http://rnacentral.org) and models were built using the B. subtilis ribosome as a template. The RNA was adjusted manually and subsequently refined using ERRASER[44]. The protein models were either generated in Phenix[45] or with Phyre2 server[46]. The obtained model was refined using Phenix real-space refinement[47] at the resolution of 5.6 Å including the simulated annealing protocol. In addition the E. coli 70S ribosome model (4YBB) was used as a reference for modeling the conserved part of the model, because it is currently resolved to the highest resolution (2.1 Å).

In order to model the densities corresponding to HPF long (185 residues), first the crystal structure of the T. thermophilus 70S ribosome bound to protein Y (YfiA) (PDB-code 4Y4O) was superposed on top of our 70S map, showing that YfiA coincides well with one patch of the density. The N-terminal domain of HPF long (residues 1–102) was placed there and subsequently refined using the original model as a guide. To model the C-terminal domain (residues 129–185) of HPF long, we threaded the L. lactis HPF long sequence based on the best fitting template from a symmetry search in COTH[48] and performed a Phyre2 search[46]. The best Phyre2 result was the model of residues 119–174 of ribosome-associated protein Y (PSrp-1) from C. acetobutylicum (PDB-code 3KA5). The most homologous structure was of Lmo2511 protein from L. monocytogenes (3K2T). Both homologs and the symmetry prediction template had the same symmetry and a highly comparable secondary structure and conserved dimer interface. The structure of PSrp-1 was used as a template for one-to-one threading, because of its annotation to protein Y (as opposed to unknown), higher resolution (1.8 vs. 2.4 Å) and the slightly better model quality. The PSrp-1 crystal structure was modeled as a dimer in the asymmetric unit.

All chains, except the HPF long C-terminal domain, were modeled to the single 5.6 Å map. The C-terminal domain of HPF long and dimer interaction of H26–H26 were modeled to the 100S map that was created from two copies of the map and started with a rigid body fit of two copies of the ribosome 70S model and the HPF long C-terminal dimer. Refinement in Phenix at 6 Å resolution yielded spontaneous base-pairing in H26 without further manual adjustments.

**Homology searches and phylogenetic tree.** To search for L. lactis HPF long homologs, its sequence (NCBI accession number WP_011834629.1) was used as a query sequence in BLASTp (http://www.ncbi.nlm.nih.gov/BLAST/), setting a limit of 10,000 (number of homologs to identify). We manually removed false hits and set additional thresholds at 30% sequence identity and 40% coverage; the latter to draw out HPF short homologs as well.

We then selected 110 homologs from as many different bacterial phyla as possible[49], choosing (if present) four sequences with full coverage and two sequences with 'half' coverage (Supplementary Fig. 10). Sequence alignment was performed by MUSCLE in MEGA7[50], using default parameters. Phylogenetic analysis was also done in MEGA7 (using the Maximum Likelihood method and the JTT matrix based model, with partial deletion of positions containing gaps and missing data). In addition, a BLASTp search with the E. coli RMF sequence was performed, yielding 711 homologs.

**Oligomeric state of HPF long.** We performed SEC-MALLS to study the oligomeric state of HPF long in vitro. MALLS is a spectroscopic technique, which has proven to be one of the best for the determination of oligomeric states of polymers, including proteins in solution[51]. For this purpose, HPF long from L. lactis encoded with N-terminal Strep-tag was overexpressed in L. lactis NZ9000 and purified on a Strep-tactin Sepharose column (IBA-GmbH, Göttingen, Germany) as described previously[6]. Purified protein was concentrated (to about 4 mg ml$^{-1}$) in buffer S (100 mM Tris-HCl, pH 8, 150 mM NaCl, 1 mM EDTA) and immediately subjected

to SEC-MALLS using Minidawn TREOS detector (Wyatt Technologies, USA) and, based on the obtained profile, the molecular mass was calculated using ASTRA software (Wyatt Technologies, USA). In total, three batches of HPF long were purified and subjected to SEC-MALLS. Two included 10% glycerol in the buffer and the final measurement was done in the absence of glycerol to flatten the baseline for the refractive index measurements. All three data sets were in agreement and showed that HPF long is a dimer.

**Data availability**. The raw data that support the findings of this study are available from the corresponding author on request. Final EM maps and model are available online under PDB accession code 5MYJ and EMDB accession code EMD-3581.

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

## Acknowledgements

We are grateful to Dr Rishi Matadeen for support at the NeCEN, Dr Laura van Bezouwen for valuable discussions regarding particle processing, Michiel Punter for his computer support and the Center for Information Technology of the University of Groningen, in particular Dr Bob Dröge, for all their computer support and for providing access to the Peregrine high-performance computing cluster. This research is financed in part by the BioSolar Cells open innovation consortium, supported by the Dutch Ministry of Economic Affairs, Agriculture and Innovation (LEF). The work was further funded by a NWO TOPGO (L.10.060), ERC Advanced grant (ABCVolume) to B.P. and NWO Vidi grant (723.014.002) to A.G.

## Author contributions

Concept supervision and design: A.G., E.J.B., and B.P.; SEC-MALLS experiments: V.A.; sample preparation: P.P.; data acquisition: G.T.O., data processing: L.E.F.; modeling and interpretation: L.E.F., T.P., and A.G.; manuscript writing: L.E.F., G.T.O., T.P., B.P., E.J.B., and A.G.

## Additional information

**Competing interests:** The authors declare no competing financial interests.

