## [Peer Review File · Nature Communications]

Reviewers' Comments:

Reviewer #1 (Remarks to the Author):

The manuscript by Franken et al. addresses a very interesting and not yet completely clear phenomenon of ribosome dimerization in bacteria during stress. This process is also referred in the literature as ribosome hibernation and plays pivotal role in survival of bacteria during harsh conditions including treatment with antibiotics and is believed to contribute to the development of drug resistance. During ribosome hibernation the individual ribosome particles are reversibly inactivated by binding to hibernation factor(s) and are stored in the cells in the translationally inactive form. There were several recently published papers uncovering mechanism of ribosome inactivation and dimerization during stress in E.coli. However, as authors pointed out, majority of bacteria have hibernation factor(s) that are different from the ones in E.coli and, therefore, could utilize different mechanism.

The current work is focused on Cryo-EM structural studies of the ribosome 100S dimers from bacterium *Lactococcus lactis* that has long version of the Hibernation Promoting Factor. Perhaps, the most interesting finding of this study is that dimerization of ribosomes from *L.lactis* occurs via different mechanism as compared to E.coli. In E.coli hibernation factors cause conformational changes of the small subunit, so that ribosomes are more prone to adhere to each other and to form dimers, while in *L.lactis* C-terminal domain of the homologous factor HPF directly mediates dimerization. Also, in this work authors reported the first cryo-EM structure of the *L.lactis* 70S ribosome, which is an important advancement of science by itself.

In my opinion, the amount of material in this study is overwhelming and is enough for at least 2, if not 3, separate papers: (i) the first structure of the 70S ribosome from *L. lactis*; (ii) novel mechanism of ribosome dimerization; (iii) phylogenetic analysis of HPF factors from various species. As a result, the manuscript turned out to be extremely long, with lots of unnecessary material and details. Also, it was prohibitively difficult to read the text, not because of the grammar, but because of the style in which the same things are re-iterated many times over again with poor logical flow. Grammar-wise the text is actually written well. While going through the text, quite often I had to re-read the same paragraph multiple times in order to get the sense. In the end, I have a strong feeling that this manuscript was initially written as several parts that were merged later.

In summary, the authors shared an interesting discovery of the alternative mechanism of

ribosome dimerization in *L.lactis*. However, in its current form the manuscript is unacceptable for publication. The main weaknesses in my opinion are (i) very poor structural organization of text and figures (too much text, too many figures and supplementary materials), and (ii) relatively low resolution of the intersubunit interface where dimerization occurs. At the same time this study provides important results and conclusions. If the main critical points denoted above and below are addressed by the authors, this manuscript could be reconsidered for publication.

Comments, suggestions and questions to the authors:

Major critical points:

1. Absence of a logical flow in the text. It seems that the authors are jumping from one topic to another and then back. The main body of the manuscript could be significantly shortened (2-3 times at least). In my opinion, authors used too much space to communicate few major findings. The main findings and conclusions of the work are very well and deeply buried in the text. The same points could be explained in much fewer words. Also there are lots of space in the results and discussion taken to explain details of how different procedures were done. It should all be moved in to Materials and Methods section. One way to significantly shorten the text would be to merge results and discussion sections, so that the results are discussed immediately after they are presented.
2. It is not very clear what is the actual resolution and what exactly are the structures that were built and are reported in this study. If you look at different classes of 100S dimers, their resolution is about 20Å. The resolution of the individual 70S monomer is around 5.6Å. What is the resolution of the contact between the two ribosomes? Based on Figure 5 it should be around 10-15Å. In any case, the resolution of the most interesting part that tentatively corresponds to the CTD of the HPF is relatively low.
3. Just as a possible suggestion, authors might consider splitting this manuscript into two: one reporting the first ribosome structure from *L.lactis* and the other one dealing with this new interesting dimerization mechanism.
4. Detailed phylogenetic analysis of bacterial HPF proteins could be actually omitted from this manuscript. It is not very much related to the dimerization mechanism and distracts a lot from the main line.

Minor comments and critical points:

1. Figure 1: In addition to pointing where L1-stalk is located, it would add more clarity if other parts of the ribosome will be labeled as well.

2. Figure 2A, B: Even to a ribosome specialist it is not clear what is the actual view in these panels. The authors might want to include insets indicating how ribosome is visualized. Also, adding labels such as “50S1”, “30S1”, “30S2” might help.
3. Figure 2C: Why there are no voxels at resolution 7A?
4. Figure 3: I would like to suggest to change the title to “Reconstruction of electron density map for ribosome dimer”. Also, from the current figure 3 and description in the text it is not very clear how exactly this reconstruction was accomplished. Making it as a scheme would be more clear. This figure could be moved to supplementary.
5. Figure 4: Panel G is actually very important for the understanding of the mechanism of dimerization and, therefore, could be made much larger.
6. Figures 7 and 8 are very much redundant and could be merged together.
7. Figure 9: What is the point of this figure? In my opinion it is drawn in the over-simplistic way. Perhaps, authors wish to schematically depict the new mechanism of dimerization? From the current cartoon it not clear at all. I think, the cartoon should be showing the new mechanism of dimerization and compare it to the previously reported mechanism from E.coli.
8. In the last sentence of the conclusions section authors stated “excellent example”. I think it is inappropriate to use this type of subjective assessment adjectives, because it is not for the authors to decide whether it is excellent or not.
9. There are no figure showing how exactly CTDs of the two HPF proteins bound to separate 70S ribosomes interact with each other, when beta strand from one CTD becomes part of the beta-sheet on the other and vice versa.

Reviewer #2 (Remarks to the Author):

The manuscript by Franken et al. presents the cryo-EM structure of dimeric 100S ribosomes from the bacterium, *Lactococcus lactis*. The authors proposed a novel mechanism of dimerization via a long version of hibernation promotion factor HPF (HPF-long, previously noted as YfiALI in Ref. 6). This mechanism is distinct from the well-studied 100S ribosome from *E.coli*, in which case the dimerization is mediated by the ribosome modulation factor (RMF). The study builds on previous reports from the co-Investigator’s lab highlighting HPF-

long protein as a candidate to both stall protein synthesis and facilitate dimer formation of the 100S ribosomes. The major finding of the paper is a 5.6Å structure of two 70S ribosomes connected by density that is attributed to the C-terminal domain of HPF-long protein. The authors provide speculation about the function of HPF-long and ribosome dimerization, particularly in an extended discussion section. Phylogenetic analysis suggests that this HPF-long mediated mechanism could be widely adopted in bacteria. The findings are interesting and the topics should attract a wide range of readers. However, the manuscript is not well written and some claims are not fully supported by the data provided. Consequently, the manuscript cannot be accepted in its current form. Below are some specific comments:

Technical issues:

- Although TEM raw micrographs or 2D averages are not included within the paper, it would seem that the dimer particles would present a serious problem for preferred orientation (side-view only).
- Related to the prior comment, the particle distribution of different views for 3D reconstruction should be included for evaluation the quality of the model.
- A full disclosure for how the density for what the authors are assigning to be the HPF-C-terminal domain is incomplete. The two sentences in the results section do not sufficiently describe how the density for this area was determined. Since this density is the primary contribution of the paper, it is important to better convince the reader that it is accurate for docking and not an artifact of imposed symmetry and/or averaging conformational heterogeneity.
- The methods indicate that the EM data were decimated four-times for image processing. Were the decimated images used throughout refinement?
- There is a concern for overfitting. At 5.6Å resolution, side-chains would not be visible. Therefore, the authors' claims of putative base-pairs within the structure are purely speculative. Similarly, Figure 4B (and 4F) should be modified to include only the c-alpha chains, as the other panels do.
- The phylogenetic analysis is not very authoritative, the threshold for defining homology is quite low (21% identity and 50% coverage).
- The authors should justify why phyre2 search selected PDB entry 3KA5 as the template instead of 3K2T, which has higher sequence identity (71% compared to 58% of 3KA5).
- It is not clear why the multi-angle light scattering data for expressed HPF-long protein was not included. These data could provide strong support for the proposed dimeric structure in solution.

Other issues that lacked clarity or require improvement:

- It is unclear how the 100S particles were formed, stabilized, and isolated. Throughout the reading, it appears that 100S formation is a cellular mechanism to stall translation under conditions of stress. However, it appears that the 100S particles are forming spontaneously in the

current system. The citation to an earlier paper is helpful, but a brief description for dimer formation in the present study is also needed.

- Regarding the claim: “After careful inspection, the purified 70S peak-fractions micrographs contained as many dimer as the 100S peak.” The authors should provide quantification to showcase that this is not due to non-specific association of 70S on the cryoEM sample grid.
- Beyond the cryo-EM structure and HPF-modeling, the overall message of the communication remains speculative and could be improved with complementary functional data. The authors cite their earlier work showing that HPF-short blocks protein synthesis and ribosome dimer formation. Although the citation is helpful and necessary, the previously published findings decrease enthusiasm for the present investigation.
- The phylogenetic study helps to develop a hypothesis for conserved or different mechanisms among bacteria, but that hypothesis is never tested. Again, the lack of additional/novel mechanistic data for HPF-long protein leaves the present communication a feeling of being incomplete.
- Despite the sub-heading in the Discussion section, the role of dimerization is never addressed. The authors identify the N-terminal domain of HPF as the culprit for blocking 70S dissociation, thereby placing translation in hibernation mode. Additional data showing a functional role for the C-terminal domain, and need for dimerization of stalled ribosomes, would increase enthusiasm for the report.
- Much of the Discussion includes unnecessary details that only add confusion to the report. The Discussion especially would be improved with more focus.
- As a similar example, it is not clear why the authors chose to include an extended evaluation of the heterogenous structures in the latter sections of the results. Much of this discussion detracts from the primary findings of the paper and should be omitted.

Response to Reviewer Comments:

Reviewer #1 (Remarks to the Author):

The manuscript by Franken et al. addresses a very interesting and not yet completely clear phenomenon of ribosome dimerization in bacteria during stress. This process is also referred in the literature as ribosome hibernation and plays pivotal role in survival of bacteria during harsh conditions including treatment with antibiotics and is believed to contribute to the development of drug resistance. During ribosome hibernation the individual ribosome particles are reversibly inactivated by binding to hibernation factor(s) and are stored in the cells in the translationally inactive form. There were several recently published papers uncovering mechanism of ribosome inactivation and dimerization during stress in E.coli. However, as authors pointed out, majority of bacteria have hibernation factor(s) that are different from the ones in E.coli and, therefore, could utilize different mechanism.

The current work is focused on Cryo-EM structural studies of the ribosome 100S dimers from bacterium *Lactococcus lactis* that has long version of the Hibernation Promoting Factor. Perhaps, the most interesting finding of this study is that dimerization of ribosomes from *L.lactis* occurs via different mechanism as compared to *E.coli*. In *E.coli* hibernation factors cause conformational changes of the small subunit, so that ribosomes are more prone to adhere to each other and to form dimers, while in *L.lactis* C-terminal domain of the homologous factor HPF directly mediates dimerization. Also, in this work authors reported the first cryo-EM structure of the *L.lactis* 70S ribosome, which is an important advancement of science by itself.

In my opinion, the amount of material in this study is overwhelming and is enough for at least 2, if not 3, separate papers: (i) the first structure of the 70S ribosome from *L. lactis*; (ii) novel mechanism of ribosome dimerization; (iii) phylogenetic analysis of HPF factors from various species. As a result, the manuscript turned out to be extremely long, with lots of unnecessary material and details. Also, it was prohibitively difficult to read the text, not because of the grammar, but because of the style in which the same things are re-iterated many times over again with poor logical flow. Grammar-wise the text is actually written well. While going through the text, quite often I had to re-read the same paragraph multiple times in order to get the sense. In the end, I have a strong feeling that this manuscript was initially written as several parts that were merged later.

In summary, the authors shared an interesting discovery of the alternative mechanism of ribosome dimerization in *L. lactis*. However, in its current form the manuscript is unacceptable for publication. The main weaknesses in my opinion are

- (i) very poor structural organization of text and figures (too much text, too many figures and supplementary materials), and

-We highly appreciate the suggestions of this reviewer and agree with his/her opinion on the structure and repetitive nature of the manuscript. We have shortened the manuscript and rewritten various sections as summarized above. Furthermore, we have modified and reduced the number of figures, and we feel that we now convey the whole story in a less convoluted manner. We rather not split the manuscript into separate papers. We prefer to keep the phylogenetic analysis as it demonstrates how widespread the here presented ribosome hibernation mechanism is (as opposed to the previously reported mechanism, which is limited to γ -Proteobacteria).

- (ii) relatively low resolution of the intersubunit interface where dimerization occurs. At the same time this study provides important results and conclusions.

-This is a valid point; however, we cannot easily improve the resolution because of the flexible nature of the dimer. Importantly, we have taken care not to over-interpret the model in this area. The density is in accordance with the shape and volume of the homologous structures (which are of similar size and fold), and there are no other un-described densities in the map that could accommodate the C-terminal domain of the dimerization factor HPF^{long}. Therefore we are confident that the C-terminal domain of HPF^{long} is to be placed at the interface. Furthermore, we now present new SEC-MALLS experiments (page5, lines 32-37) to demonstrate unambiguously that HPF^{long} is a dimer in solution (Figure 5), which reinforces the mechanism we present in figure 8.

If the main critical points denoted above and below are addressed by the authors, this manuscript could be reconsidered for publication.

Comments, suggestions and questions to the authors:

Major critical points:

1. Absence of a logical flow in the text. It seems that the authors are jumping from one topic to another and then back. The main body of the manuscript could be significantly shortened (2-3 times at least). In my opinion, authors used too much space to communicate few major findings. The main findings and conclusions of the work are very well and deeply buried in the text. The same points could be explained in much fewer words. Also there are lots of space in the results and discussion taken to explain details of how different procedures were done. It should all be moved in to Materials and Methods section. One way to significantly shorten the text would be to merge results and discussion sections, so that the results are discussed immediately after they are presented.

-We agree with the reviewer and have shortened the text significantly (see summary of textual changes); the Results and Discussion Sections are now each 3.5 pages, but we prefer to keep them separate, to ensure logical flow of text and prevent repetition.

2. It is not very clear what is the actual resolution and what exactly are the structures that were built and are reported in this study. If you look at different classes of 100S dimers, their resolution is about 20Å. The resolution of the individual 70S monomer is around 5.6Å. What is the resolution of the contact between the two ribosomes? Based on Figure 5 it should be around 10-15Å. In any case, the resolution of the most interesting part that tentatively corresponds to the CTD of the HPF is relatively low.

- The resolution at the interface is around 6-10Å, but this did not prevent the assignment of its density to the dimeric C-terminal domain of HPF^{long} and the modelling of its main chain. This was also possible due to the fact that two homologous structures with the same dimer interface and fold were present. Their structures already described the C-terminal dimer of HPF^{long} very well. Because of these models we are confident that our assignment of the center density to the HPF^{long} C-terminal domains is correct, and that our proposed mechanism that follows from this result is valid.

3. Just as a possible suggestion, authors might consider splitting this manuscript into two: one reporting the first ribosome structure from *L.lactis* and the other one dealing with this new interesting dimerization mechanism.

-We appreciate this suggestion, but feel that reporting the first ribosome structure from *L. lactis* alone would not justify publication of the mechanism in Nature Communications. The new dimerization mechanism is the most important finding of our research, which builds on the ribosome structure from *L. lactis*. We thus prefer to keep both stories connected in one paper.

4. Detailed phylogenetic analysis of bacterial HPF proteins could be actually omitted from this manuscript. It is not very much related to the dimerization mechanism and distracts a lot from the main line.

-We feel that the phylogenetic analysis underlines the relevance of the new dimerization mechanism (broader impact), but we have shortened the text and made the analysis more robust in accordance with the comments of reviewer #2.

Minor comments and critical points:

1. Figure 1: In addition to pointing where L1-stalk is located, it would add more clarity if other parts of the ribosome will be labeled as well.

-For overview, we now labeled 50S and 30S. We are hesitant in adding more labels as the major function of this figure is to show the rotation in different conformations, for which the L-stalk is a very nice indicator. To make this more visible we colored the L1 stalks.

2. Figure 2A, B: Even to a ribosome specialist it is not clear what is the actual view in these panels. The authors might want to include insets indicating how ribosome is visualized. Also, adding labels such as "50S1", "30S1", "30S2" might help.

- To make the figure clearer we added the low-resolution map of panel A from figure 1 in the background and adjusted the figure subscript to explain more clearly that panel A and B represent the same view. Figure 2 now also contains the 50S and 30S indications, and for clarity we marked two very distinct ribosomal areas (16S-H6 and L1 stalk) in Figure 2.

3. Figure 2C: Why there are no voxels at resolution 7A?

-This is indeed surprising. Resmap is a program that gives an approximation of the resolution rather than exact numbers, but we do not have a good explanation for this finding. The color-scheme in panels A and B, which originates from the same program, does include voxels at a resolution of 7A, and we therefore decided to delete Panel C.

4. Figure 3: I would like to suggest to change the title to "Reconstruction of electron density map for ribosome dimer". Also, from the current figure 3 and description in the text it is not very clear how exactly this reconstruction was accomplished. Making it as a scheme would be more clear. This figure could be moved to supplementary.

-We changed the title to "Reconstruction of the 100S density map for the ribosome dimer". We rewrote this part and moved the text to the materials and methods section (page 12, lines 27-32) and the improved figure to the supplement (S6), where it is also described more clearly now.

5. Figure 4: Panel G is actually very important for the understanding of the mechanism of dimerization and, therefore, could be made much larger.

- On the basis of suggestions from reviewer 2, we removed panel F. Panel G was expanded and became panel F in the revised manuscript.

6. Figures 7 and 8 are very much redundant and could be merged together.

-We moved Figure 7 to the supplementary information.

7. Figure 9: What is the point of this figure? In my opinion it is drawn in the over-simplistic way. Perhaps, authors wish to schematically depict the new mechanism of dimerization? From the current cartoon it not clear at all. I think, the cartoon should be showing the new mechanism of dimerization and compare it to the previously reported mechanism from *E.coli*.

-We agree with the reviewer. Therefore, we have removed this figure and replaced it with a new schematic that shows the main differences between the two mechanisms (Figure 8).

8. In the last sentence of the conclusions section authors stated “excellent example”. I think it is inappropriate to use this type of subjective assessment adjectives, because it is not for the authors to decide whether it is excellent or not.

-We have removed the subjective statement.

9. There are no figure showing how exactly CTDs of the two HPF proteins bound to separate 70S ribosomes interact with each other, when beta strand from one CTD becomes part of the beta-sheet on the other and vice versa.

-If we understand correctly, the reviewer would like to see the dimerization mechanism through binding of two copies of HPF first to the ribosome and then to each other? This is however not how we envisage the dimerization of the ribosomes. Instead, HPF is already dimeric in solution (see new figure 5) and binds two monomeric ribosomes consecutively. We have enlarged figure 4G, which shows better the beta-sheets and rewritten the discussion. The new figure 8 explains more clearly the new dimerization mechanism and how it differs from that of *E. coli*.

Reviewer #2 (Remarks to the Author):

The manuscript by Franken et al. presents the cryo-EM structure of dimeric 100S ribosomes from the bacterium, *Lactococcus lactis*. The authors proposed a novel mechanism of dimerization via a long version of hibernation promotion factor HPF (HPF-long, previously noted as YfiAL1 in Ref. 6). This mechanism is distinct from the well-studied 100S ribosome from *E.coli*, in which case the dimerization is mediated by the ribosome modulation factor (RMF). The study builds on previous reports from the co-Investigator's lab highlighting HPF-long protein as a candidate to both stall protein synthesis and facilitate dimer formation of the 100S ribosomes. The major finding of the paper is a 5.6Å structure of two 70S ribosomes connected by density that is attributed to the C-terminal domain of HPF-long protein. The authors provide speculation about the function of HPF-long and ribosome dimerization, particularly in an extended discussion section. Phylogenetic analysis suggests that this HPF-long mediated mechanism could be widely adopted in bacteria. The findings are interesting and the topics should attract a wide range of readers. However, the manuscript is not well written and some claims are not fully supported by the data provided. Consequently, the manuscript cannot be accepted in its current form. Below are some specific comments:

-We are glad that this reviewer found the reported results interesting, and we agree that the manuscript was not written in the most optimal way. We are also grateful for the comments and suggestions. We have modified the manuscript as summarized above; see also our response to the comments of reviewer 1.

Technical issues:

- Although TEM raw micrographs or 2D averages are not included within the paper, it would seem that the dimer particles would present a serious problem for preferred orientation (side-view only).

- Side-views are indeed much more abundant, but this is not a problem as we have good coverage in angles. This is now illustrated with supplementary figure 2.

- Related to the prior comment, the particle distribution of different views for 3D reconstruction should be included for evaluation the quality of the model.

-We have added supplementary figure 2, which shows the 2D projections and angular distributions, and a reference to this figure is given in the figure caption of figure 2.

- A full disclosure for how the density for what the authors are assigning to be the HPF-C-terminal domain is incomplete. The two sentences in the results section do not sufficiently describe how the density for this area was determined. Since this density is the primary contribution of the paper, it is important to better convince the reader that it is accurate for docking and not an artifact of imposed symmetry and/or averaging conformational heterogeneity.

-The reconstruction of the entire 100S particle was improved by fitting two copies of the monomeric structure into the map of lower resolution and subsequently into each other. There is a high correlation between the two copies, which is shown for the central area in what is now supplementary figure 6C and D. A close look at the better known protein uS2, which is also in the center of the dimer, shows that there is no negative effect of this procedure. If this would introduce artefacts or if there would be conformational heterogeneity, it would deform or disturb this density as well and this is not the case. The procedure is now better described (page 12, lines 27-32) and more extensive in the revised supplementary figure 6.

- The methods indicate that the EM data were decimated four-times for image processing. Were the decimated images used throughout refinement?

-No this is only true for the initial classes in figure 1. We have stated this more clearly now (page 12, line 13).

- There is a concern for overfitting. At 5.6Å resolution, side-chains would not be visible. Therefore, the authors' claims of putative base-pairs within the structure are purely speculative. Similarly, Figure 4B (and 4F) should be modified to include only the c-alpha chains, as the other panels do.

-We appreciate the reviewers' concern. However, for many side-chains, particularly for the larger ones, we see some density in the map. Furthermore, refinement without side-chains produced somewhat worse values of Correlation Coefficient of our model to fit the map. Figure 4F showed side-chains that we cannot assign unambiguously, but these were shown for their hydrophobic nature and were partly based on their location in the homology structures. We now describe this data in the text and have replaced panel F in figure 4 with panel G.

The last point raised is that the base-pair interaction of H26 with H26 is speculative. While indeed the resolution would not be good enough to see these individual base-pairing, rigid-body fitting of the two 70S models into the 100S map places the complementary base-pairs exactly opposite of each other. Subsequent refinement in Phenix causes both loops to open and yields the base-pairing without any manual interference or pre-modeling (page 13, lines 20-21). Also, this conformation is extremely stable, considering the lack of floppiness in the EM data. We thus believe that this is an extremely likely scenario, but we have toned down the text (page 5, line 9) as implicitly suggested by the reviewer.

- The phylogenetic analysis is not very authoritative, the threshold for defining homology is quite low (21% identity and 50% coverage).

We checked for incorrect hits (~300 out of 10000) and removed them from the sequence set. Furthermore, we have adjusted our search putting two additional thresholds 30% identity and a coverage of 40%, leaving a total of ~7000 homologues. The reason we allowed the coverage to go to 40% is because at this coverage the HPF^{short} sequences will be drawn out of the database. We adjusted the results (page 6, lines 2-4) of the revised manuscript and the methods (page 13, lines 24-28).

- The authors should justify why phyre2 search selected PDB entry 3KA5 as the template instead of 3K2T, which has higher sequence identity (71% compared to 58% of 3KA5).

3K2T and 3KA5 have highly similar secondary structures and dimeric structures and interfaces and are thus interchangeable. The main difference is that the 3K2T model has a gap in the structure and 3KA5 has a better resolution (1.8 Å vs 2.4 Å). Phyre indicates 3KA5 as slightly more reliable, and, since the model is better connected, we decided to use 3KA5. We have added this information (page 13, lines 9-16).

- It is not clear why the multi-angle light scattering data for expressed HPF-long protein was not included. These data could provide strong support for the proposed dimeric structure in solution.

We agree with the reviewer that this is a very important experiment and we have now included it in the manuscript (page 5, lines 32-37; and figure 5)

Other issues that lacked clarity or require improvement:

- It is unclear how the 100S particles were formed, stabilized, and isolated. Throughout the reading, it appears that 100S formation is a cellular mechanism to stall translation under conditions of stress. However, it appears that the 100S particles are forming spontaneously in the current system. The citation to an earlier paper is helpful, but a brief description for dimer formation in the present study is also needed.

We now explain better in the discussion (page 9 lines 18-25) how co-purification of dimeric HPF^{long}, bound to monomeric ribosomes, leads to 100S formation. Since HPF^{long} is always present as dimer, it will scavenge monomeric ribosomes and ultimately form dimeric ribosomes (see also our new figure 8)

- Regarding the claim: "After careful inspection, the purified 70S peak-fractions micrographs contained as many dimer as the 100S peak." The authors should provide quantification to showcase that this is not due to non-specific association of 70S on the cryoEM sample grid.

We used 2D and 3D classification to remove non-specific ribosome oligomers from the dataset. The final map still contains many dimers from the monomeric dataset (14,782 from the monomeric and 28,747 from the dimeric peak fractions), showing that it is not just coincidental and that there is no distinctive difference. Besides the particle processing, also our mechanistic model justifies the combining of the two datasets. In fact, with dimeric HPF^{long} one expects the ribosomes to form dimers when e.g. mRNA and charged-tRNAs are limiting. We have modified the text to make this point clearer in the discussion (page 9, lines 18-25) and the methods (page 11, lines 25-29).

- Beyond the cryo-EM structure and HPF-modeling, the overall message of the communication remains speculative and could be improved with complementary functional data. The authors cite their earlier work showing that HPF-short blocks protein synthesis and ribosome dimer formation. Although the citation is helpful and necessary, the previously published findings decrease enthusiasm for the present investigation.

The focus of the current manuscript is on the mechanism and structural basis for ribosome dimerization, which is very different from our previous physiological work. Moreover, the conformational change that is suggested by negative staining of the ribosomes is now shown to be incorrect, advancing our understanding; the here-presented cryo-TEM studies now reveal for the first time the mechanism of ribosome dimerization that is relevant for the majority of bacterial species known to date. We have rewritten the Discussion section and removed most of the redundancy when compared to our previous work.

- The phylogenetic study helps to develop a hypothesis for conserved or different mechanisms among bacteria, but that hypothesis is never tested. Again, the lack of additional/novel mechanistic data for HPF-long protein leaves the present communication a feeling of being incomplete.

The phylogenetic study is indeed mainly supporting the importance of this new mechanism with respect to the known mechanism and placing it in evolutionary perspective, while also identifying candidates for future research. It shows that the majority of bacterial species have hibernation factors that correspond to HPF^{long} rather than HPF^{short} in combination with RMF. It also shows that the majority of HPF^{long} homologues have a linker between the N- and C-domain that is long enough to connect each N-terminal domain to each C-terminal domain (Figure 3). We feel that this information is compelling evidence for the proposed mechanism, even without testing on ribosomes from other species, which is currently not possible as it is unfortunately needed to have the ribosome dimeric structures of other species for detailed comparisons (which clearly is beyond the scope of this paper).

- Despite the sub-heading in the Discussion section, the role of dimerization is never addressed. The authors identify the N-terminal domain of HPF as the culprit for blocking 70S dissociation, thereby placing translation in hibernation mode. Additional data showing a functional role for the C-terminal domain, and need for dimerization of stalled ribosomes, would increase enthusiasm for the report.

We feel that we address this point, but we have made it more clear in the revised manuscript. What the reviewer proposes has to a large extent been addressed in our previous work (Puri, P. *et al.* 2014. *Mol. Microbiol.* **91**, 394-407). We have shown that deletion of the C-terminal domain prevents the protein from ribosome dimerization, and the corresponding strains are affected in their growth (recovery from the stationary growth phase). We now present the molecular basis for these physiological observations.

- Much of the Discussion includes unnecessary details that only add confusion to the report. The Discussion especially would be improved with more focus.

We agree with the reviewer and have shortened and better focused the Discussion section.

- As a similar example, it is not clear why the authors chose to include an extended evaluation of the heterogeneous structures in the latter sections of the results. Much of this discussion detracts from the primary findings of the paper and should be omitted.

If we are not mistaken the reviewer means the rigid body fitting of the ribosome model in the lower resolution. We agree on this point and have shortened this part of the text (page 5 lines 22-28) and removed the last three panels from the figure 4.

Reviewers' Comments:

Reviewer #1 (Remarks to the Author):

After reading the revised version of the manuscript, I need to say that the authors did a good job revising the manuscript. Good, but not excellent! The manuscript definitely became clearer after the revision, but I would like to kindly insist on few more changes. Although it is absolutely fine that the authors preferred not to follow some of the suggestions, I think the manuscript still suffers some problems that need to be addressed:

1. Although the authors did shorten "Results" and "Discussion" sections by moving the methodological parts to the more appropriate "Methods" section, I still think that the manuscript is prohibitively long to be read by most of the readers! The "Discussion" section still occupies more than three full pages of continuous text, which I find way too much for relatively few major findings to be discussed: (1) structure of the 70S ribosome from new species *Lactococcus lactis*; (2) novel mechanism of ribosome dimerization; (3) phylogenetic analysis of HPF factors from various species. These days, only a rare reader actually reads papers from the beginning to the end - usually readers read the abstract, quickly look through the figures and selected parts of the main text that they got interested in. Therefore, the authors should aim at delivering their results in a very concise and clear form, without multiple iterations of the same facts.

2. I find figure 1 very misleading! In my opinion, the figures should be self-explanatory without the necessity to read the legend. My first impression of this figure was that it shows electron density for the 100S dimer in three different orientations, however, this figure actually depicts 3 out of 9 classes of particles. The authors might wish to move this figure to the supplementary, since in most of the cryo-EM papers on the ribosome, this type of figures are usually included in the supplementary. This figure, does not really carry any important information. Also, authors should expand this figure and show all nine classes of their particles with the corresponding percentages. The authors should also show directly on the figure which class was used for model fitting.

In my opinion, Figure 1 is one of the most important and precious spots in the whole paper, where all readers always look, therefore authors might wish to present their most important and valuable result here.

3. As I pointed out in my original review, one of the important discoveries reported in this study is the first structure of the 70S ribosome from *Lactococcus lactis*. Therefore, this important result should be communicated to the reader by moving Figure S5 from the Supplementary to the main text, while moving and expanding Figure 1 from the main text to the supplementary (see previous comment).

4. Figure 2 should also be moved from the main text to the supplementary for the same reason as current figure 1 - it has no conceptual value and simply describes the quality of the structure and degree of conformational freedom in the dimer.

5. The new figure 5 is not self-explanatory. Molecular masses should be indicated on it and also the molecular mass of the HPF monomer should be indicated - so that by looking at the figure it will be immediately clear to the reader that in solution HPF exists as a dimeric and not monomeric form.

6. I haven't noticed this before, but in Figure 6 *Thermus* is annotated as Gram-positive bacterium, while it should be Gram-negative.

Reviewer #2 (Remarks to the Author):

The revised manuscript is a much improved version of the original submission. However, many issues remain unclear and should be better addressed before publication. Some minor comments are also included below.

1) It seems that 100S particles are naturally occurring complexes that coincide with stationary phase bacteria and are separated (without induction, stress, etc) from 70S monomers on a sucrose gradient. However, upon reading the citation for complex formation (ref. 6) it appears that the 100S particles are formed upon overexpression of HPF-long. It is important to clarify how the 100S particles are being formed, as this would help the reader understand that the 70S-70S interaction is due to HPF-long exclusively (and not another factor such as RMF).

2) It would help in defining the biological significance of dimerization to report the relative ratio of 100S to 70S particles that are isolated under the conditions being used. It would also help to report the relative ratio of 70S:100S under stationary versus log phase (when HPF is not overexpressed).

3) the conclusion (figure 8) claiming that HPF-long can gather one, then two ribosomes in solution is not clear. There is no evidence presented of HPF-long binding to a 70S ribosome. The way the paper is written it is difficult to tell whether this model is proposed or being presented as data. It would be helpful to clarify which is the case.

4) The authors should make every effort to be completely accurate in the communication. As one example, pg. 9 lines 9-10 still states that H26 sequence facilitates base-pairing. However (as stated in the prior critique) the base-pairing model is only speculative and should be clarified.

The authors include that the H26 sequence is not conserved among species, which argues against a H26-H26 base-pair. It is more likely that the two helices merely come into proximity in one of the reconstructions (and not the other).

5) Pg. 2 line 11-12: It is not understood what is meant by HPF binding to 90S particles to form mature 100S particles. 'Mature' is probably not the correct word here as it suggests a step in biogenesis. Also, it would help to clarify what the 90S dimer is and how it differs from the 70S-70S dimeric 100S particle.

6) While the revised paper is a definite improvement in organization and clarity, the discussion of the revised manuscript remains difficult to follow.

7) The Conclusion section also appears to be more speculation than actual findings. The paper presents the structure of *L. lactis* 100S ribosomes and focuses on the HPF-long interactions. However, despite the focus of the conclusion section, the present study never attempts to address stoichiometry of HPF-long with 70S ribosomes, nor does it compare expression levels of HPF-long in *Lactis* or RMF in *Coli*.

Response to Reviewer Comments:

Reviewers' comments:

Reviewer #1 (Remarks to the Author):

After reading the revised version of the manuscript, I need to say that the authors did a good job revising the manuscript. Good, but not excellent! The manuscript definitely became clearer after the revision, but I would like to kindly insist on few more changes. Although it is absolutely fine that the authors preferred not to follow some of the suggestions, I think the manuscript still suffers some problems that need to be addressed:

1. Although the authors did shorten "Results" and "Discussion" sections by moving the methodological parts to the more appropriate "Methods" section, I still think that the manuscript is prohibitively long to be read by most of the readers! The "Discussion" section still occupies more than three full pages of continuous text, which I find way too much for relatively few major findings to be discussed: (1) structure of the 70S ribosome from new species *Lactococcus lactis*; (2) novel mechanism of ribosome dimerization; (3) phylogenetic analysis of HPF factors from various species. These days, only a rare reader actually reads papers from the beginning to the end - usually readers read the abstract, quickly look through the figures and selected parts of the main text that they got interested in. Therefore, the authors should aim at delivering their results in a very concise and clear form, without multiple iterations of the same facts.

Reply: In line with the suggestion of the reviewer, we have further shortened the discussion to about two pages. We now focus on the three main findings as pointed out by this reviewer.

2. I find figure 1 very misleading! In my opinion, the figures should be self-explanatory without the necessity to read the legend. My first impression of this figure was that it shows electron density for the 100S dimer in three different orientations, however, this figure actually depicts 3 out of 9 classes of particles. The authors might wish to move this figure to the supplementary, since in most of the cryo-EM papers on the ribosome, this type of figures are usually included in the supplementary. This figure, does not really carry any important information. Also, authors should expand this figure and show all nine classes of their particles with the corresponding percentages. The authors should also show directly on the figure which class was used for model fitting.

In my opinion, Figure 1 is one of the most important and precious spots in the whole paper, where all readers always look, therefore authors might wish to present their most important and valuable result here.

Reply: The reviewer may have misunderstood our text, for which we apologize. All nine classes belong to one of the conformations shown in the Supplementary Figure 1, and the percentages therefore cover the complete clean dataset. There are no extra figures to display. As suggested by reviewer, we moved Figure 1 and 2 to the supplement.

3. As I pointed out in my original review, one of the important discoveries reported in this study is the first structure of the 70S ribosome from *Lactococcus lactis*. Therefore, this important result should be communicated to the reader by moving Figure S5 from the Supplementary to the main text, while moving and expanding Figure 1 from the main text to the supplementary (see previous comment).

Reply: We agree with that and now Figure S5 is the new Figure 1.

4. Figure 2 should also be moved from the main text to the supplementary for the same reason as current figure 1 - it has no conceptual value and simply describes the quality of the structure and degree of conformational freedom in the dimer.

Reply: Done.

5. The new figure 5 is not self-explanatory. Molecular masses should be indicated on it and also the molecular mass of the HPF monomer should be indicated - so that by looking at the figure it will be immediately clear to the reader that in solution HPF exists as a dimeric and not monomeric form.

Reply: This figure has been modified and now the calculated mass is shown. We do not however show in the figure the mass of the monomer since it is uncommon and usually only the calculated mass shown.

6. I haven't noticed this before, but in Figure 6 Thermus is annotated as Gram-positive bacterium, while it should be Gram-negative.

Reply: corrected.

Reviewer #2 (Remarks to the Author):

The revised manuscript is a much improved version of the original submission. However, many issues remain unclear and should be better addressed before publication. Some minor comments are also included below.

1) It seems that 100S particles are naturally occurring complexes that coincide with stationary phase bacteria and are separated (without induction, stress, etc) from 70S monomers on a sucrose gradient. However, upon reading the citation for complex formation (ref. 6) it appears that the 100S particles are formed upon overexpression of HPF-long. It is important to clarify how the 100S particles are being formed, as this would help the reader understand that the 70S-70S interaction is due to HPF-long exclusively (and not another factor such as RMF).

Reply: We have rewritten the discussion section but also significantly shortened the original text. In brief, HPF^{long} is a dimeric protein and can bind to 70S and either or not recruit a second ribosome and thus form 100S ribosomes. Our findings indicate that the 70S-70S interaction is due to HPF^{long} exclusively. With increasing expression of HPF^{long}, more 100S ribosomes are formed but there will be an optimum. If HPF^{long} is produced in excess of ribosomes, the equilibrium will shift from 100S to 70S (with dimeric HPF^{long} bound to monomeric ribosomes). Next, to the HPF^{long}/ribosome ratio the presence of mRNA (more abundant in the exponential than in the stationary phase) plays a role. We feel that the new text and the model in figure 7 clarify this point.

2) It would help in defining the biological significance of dimerization to report the relative ratio of 100S to 70S particles that are isolated under the conditions being used. It would also help to report the relative ratio of 70S:100S under stationary versus log phase (when HPF is not overexpressed).

Reply: See our response to point one. We have no quantitative data on the 70S:100S ratio. This is very hard to do properly because preparations for EM (e.g. sucrose gradient centrifugation; dialysis to remove the sucrose) take time and this affects the ratio of monomeric over dimeric ribosomes.

3) the conclusion (figure 8) claiming that HPF-long can gather one, then two ribosomes in solution is not clear. There is no evidence presented of HPF-long binding to a 70S ribosome. The way the paper is written it is difficult to tell whether this model is proposed or being presented as data. It would be helpful to clarify which is the case.

Reply: 70S ribosomes and HPF^{long} copurify, see also Ueta *et al.* 2010. This paper also shows that the ratio between 100S-HPF^{long} and 70S-HPF^{long} is not constant and dependent on the stage of growth. So, our model is based on the structural studies present in this paper, our earlier physiological studies (Puri *et al.* 2014) and the work of Ueta *et al.* 2010. We now make this more clear in the revised manuscript.

4) The authors should make every effort to be completely accurate in the communication. As one example, pg. 9 lines 9-10 still states that H26 sequence facilitates base-pairing. However (as stated in the prior critique) the base-pairing model is only speculative and should be clarified. The authors include that the H26 sequence is not conserved among species, which argues against a H26-H26 base-pair. It is more likely that the two helices merely come into proximity in one of the reconstructions (and not the other).

Reply: We think we have strong evidence for the base-pairing, but in order to satisfy the reviewer we have removed the corresponding sentence.

5) Pg. 2 line 11-12: It is not understood what is meant by HPF binding to 90S particles to form mature 100S particles. 'Mature' is probably not the correct word here as it suggests a step in biogenesis. Also, it would help to clarify what the 90S dimer is and how it differs from the 70S-70S dimeric 100S particle.

Reply: 90S ribosomes have RMF but lack HPF^{short}. We have removed the word 'mature'.

6) While the revised paper is a definite improvement in organization and clarity, the discussion of the revised manuscript remains difficult to follow.

Reply: We have further streamlined the discussion, see also response to comment #1 of Reviewer#1.

7) The Conclusion section also appears to be more speculation than actual findings. The paper presents the structure of *L. lactis* 100S ribosomes and focuses on the HPF-long interactions. However, despite the focus of the conclusion section, the present study never attempts to address stoichiometry of HPF-long with 70S ribosomes, nor does it compare expression levels of HPF-long in *lactis* or RMF in *coli*.

Reply: The reviewer is partly right; we combine our own findings with data already reported in the literature. We prefer to keep the Conclusion section in the present form, now that the Discussion has been shortened to two pages.

Reviewers' Comments:

Reviewer #2 (Remarks to the Author):

The authors have sufficiently addressed all concerns that arose from the prior submissions. In this reviewer's opinion, the revised article is better communicated and interpreted and the results would be of general interest to the readers of Nature Communications.